# LABEL-WISE UNCERTAINTY DECOMPOSITION FOR MULTI-LABEL CLASSIFICATION BY MAXIMIZING TYPE II LIKELIHOOD

## ABSTRACT

Currently, the way deep learning models recognize uncertainty remains inconsistent with human perception. In multi-label classification, quantifying uncertainty at the label level presents challenges, as each label may exhibit distinct model confidence levels. Understanding and decomposing label-specific uncertainty is essential for interpreting model behavior and ensuring reliable predictions. We build a hierarchical Bayesian methodology for multi-label classification that leverages a Type II likelihood and Empirical Bayes. Then we estimate and decompose label-wise uncertainties by the bias-variance decomposition. Our approaches offer four main contributions: (1) it is data centric, as Type II likelihood maximization ensures higher data likelihood; (2) it decomposes label-wise uncertainty into the model variance, the model bias and data noise; (3) our uncertainty represented by model bias is intuitively interpretable when combined with observational data; and (4) when applied to out-of-distribution (OOD) detection task, it achieves a $6.88\%$ lower FPR95 score on NUS-WIDE.

## 1 INTRODUCTION

The growing adoption of deterministic neural networks (NN) in safety-critical domains, such as medical diagnosis and autonomous driving, necessitates the development of reliable methods for uncertainty quantification. Model confidence is also essential for understanding the model's behavior, enabling interpretable predictions and informed decision-making in complex domains. When considering multi-label classification tasks, this challenge becomes even more pronounced as the model must simultaneously predict multiple labels that may exhibit interdependencies, varying levels of noise associated with different labels, and differing degrees of uncertainty across instances.

Unlike single-label multi-class classification, where each instance is assigned one label, multi-label classification allows multiple labels to exist simultaneously per instance. Consequently, individual labels can exhibit varying levels of uncertainty arising from input-dependent noise or intricate interdependencies. For example, in an image classification task, a stable foreground person may be assigned a label with low uncertainty due to clear visual features, while a moving car in the background, blurred by motion, may yield a label with high uncertainty due to ambiguity in its detection. Understanding label-wise uncertainty is crucial for interpreting the model's reasoning, as it reveals how confidently the model assigns each label and highlights potential ambiguities in its predictions. However, merely making simple use of probability is often not enough.

Adapting uncertainty estimation techniques to multi-label classification remains challenging. While Wang et al. (2021) proposed a label-wise uncertainty measurement based on an energy function, a method that leverages the overconfidence (Nguyen et al., 2015) of neural networks on out-of-distribution data. Their method primarily targets OOD detection and does not explicitly capture a deeper model uncertainty. Sale et al. (2024) discussed how the mutual information or the total variance decomposition can be used to find the label-wise uncertainty in a multi-class classification problem. The Gibbs classifier is commonly adopted in EDL research (Sensoy et al., 2018; Zhao et al., 2023; Charpentier et al., 2020). And the Subjective Logic (SL) (Jsang, 2018) from evidence theories also plays a crucial role in defining the epistemic uncertainty (Hüllermeier & Waegeman,

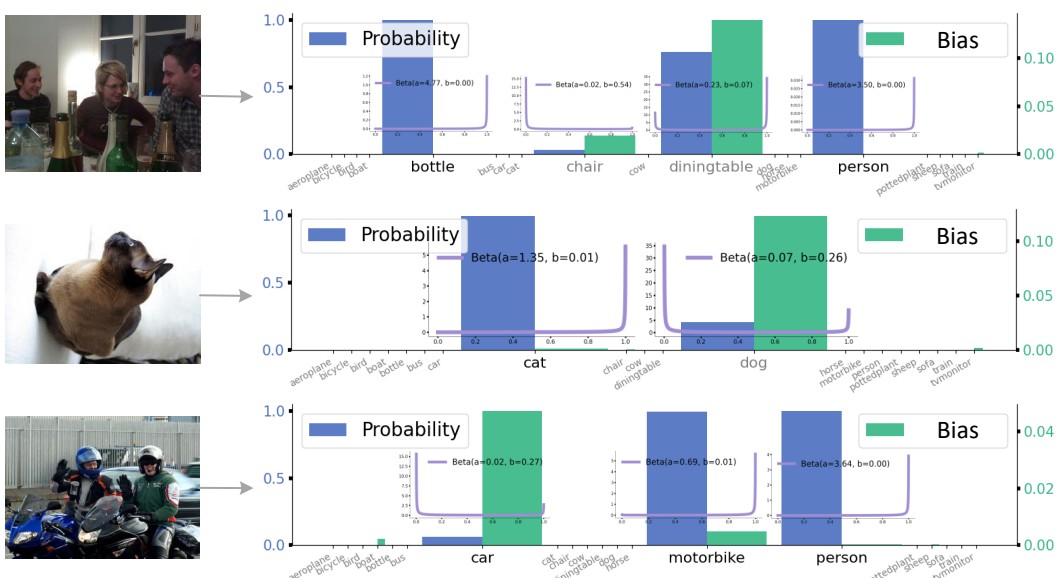

Figure 1: Given an input, the neural network is trained to predict the hyperparameters of a Type II likelihood instead of directly outputting the probability. The black labels represent the ground truth labels. The purple distribution represents the posterior Beta distribution for each label. The blue bar indicates predictive probability of the label, and the green bar represents the model bias. Across these three examples, our model demonstrates interpretable uncertainty regarding the labels: "dining table," visible but not ground truth; "dog," likely misidentified as a dog; and "car," exhibiting motion blur. This aligns well with human visual understanding of the corresponding scenes.

2021) of EDL research. As a sampling-based expected loss classifier, the Gibbs classifier does not fully optimize the probability of the observed data under the model, which is discussed in Section 2.4.

A key aspect of Bayesian inference involves assigning prior distributions to hierarchical models to quantify and estimate uncertainty Gelman (2006). Sensoy et al. (2018) introduces Evidential Deep Learning methods, a framework for modeling predictive uncertainty in multi-class classification problems by parameterizing a Dirichlet prior over first-order distribution. Malinin & Gales (2018) proposed a new framework for modeling predictive uncertainty called Prior Networks, which explicitly models distributional uncertainty. Gaussian distribution is commonly used to model and capture heteroscedastic uncertainty (Lakshminarayanan et al., 2017). The paper Nix & Weigend (1994) addressed regression problems by modeling the observed value as a sample from a heteroscedastic Gaussian distribution. Building on Gaussian, Amini et al. (2020) introduced a hierarchical evidential framework by incorporating an additional Gaussian and an inverse-gamma distribution to govern the original Gaussian's mean and variance. However, these approaches could not capture label-wise uncertainty, and some are still controversial. Bengs et al. (2022); Shen et al. (2024); Meinert et al. (2023) showed that some evidential uncertainty methods do not incentivize the learner to represent epistemic uncertainty in a faithful way. Hence, further exploration of the remaining gaps in evidential learning is essential. In this paper, we systematically build a hierarchical model for multi-label classification by Bayesian inference and theoretically prove its correctness as both a learning criterion and uncertainty measurement.

1. We define a Type II likelihood for multi-label classification and propose a maximizing Type II likelihood learning criterion. This criterion is theoretically proven to be data-centric.

2. We propose a methodology for decomposing uncertainty at the label level, which is directly tied to the definition of the Type II likelihood.

3. Experiments show that the model bias from the decomposition is more intuitively interpretable. And its statistical effectiveness is proven by label-wise OOD detection tasks.

As illustrated in Figure 1, given an input, the neural network is trained to predict the parameters of a Type II likelihood or marginal likelihood. In the context of the first image depicting a party scene,

the model shows uncertainty about the labels for the chair and dining table. Notably a portion of the dining table is visible in this image but not treated as ground truth.

## 2 METHODOLOGY

In multi-label classification, a sample can be associated with multiple labels simultaneously, requiring the prediction of a binary label vector $\mathbf{y} = (y_1, y_2, \ldots, y_K)^\top \in \{0, 1\}^K$ from input feature $\mathbf{x} \in \mathbb{R}^d$. The objective is to learn a function $f$ such that $f(\theta, \mathbf{x})$ helps approximate a probability vector $M = [\mu_1, \mu_2, \ldots, \mu_K]$, where $\mu_k$ represents the probability of the $k$-th label of the input $\mathbf{x}$.

In this work, we focus on the Type II Maximum Likelihood (ML) methodology. While Type I ML is simpler and computationally efficient due to its deterministic nature, Type II ML provides a more comprehensive framework for capturing and exploiting uncertainty. Further introduction on both Type I and Type II ML approaches can be found in Appendix A.

### 2.1 TYPE I MAXIMUM LIKELIHOOD OF MULTI-LABEL CLASSIFICATION

The binary cross-entropy loss Rumelhart et al. (1986) function is used for multi-label classification tasks due to the independent nature of the problem. It is particularly effective as it treats each label as a separate binary classification, allowing the model to independently evaluate the presence or absence of each label. Let $y_{ik}$ be the ground truth label for the $i$-th sample ($y_{ik} \in \{0, 1\}$) and $k$-th label. $\hat{y}_{ik}$ represents the raw logit output of the model $f$ for the $i$-th input $\mathbf{x}_i$. $\sigma(\hat{y}_{ik}) = \frac{1}{1+e^{-\hat{y}_{ik}}}$ is the sigmoid function applied to the logits, converting them to probabilities $\mu$. $N$ denotes the total number of samples. The binary cross-entropy empirical risk using the sigmoid function can be defined as:

$$\hat{\mathcal{R}}_{\mathcal{D}}^{\text{BCE}}(f(\theta)) = -\frac{1}{NK} \sum_{i=1}^{N} \sum_{k=1}^{K} \left[ y_{ik} \cdot \log\left(\sigma(\hat{y}_{ik})\right) + (1 - y_{ik}) \cdot \log\left(1 - \sigma(\hat{y}_{ik})\right) \right],$$

where is derived from the negative log-likelihood of a Bernoulli distribution, and the target is to maximize the Type I likelihood of the observed data. For a single label, the likelihood is given by:

$$p(y \mid \mu) = \mu^y (1 - \mu)^{1-y}, \tag{1}$$

where $\mu \in [0, 1]$ represents the predicted probability of the positive label $y = 1$.

In Type I ML, the hyperparameter $\mu$ is assumed as highly peaked, and uncertainty is often represented through the spread of the predicted probabilities, Entropy, etc. However, these methods only capture the uncertainty based on the probability values themselves. It cannot account for deeper levels of uncertainty, such as cases where predictions with the same probability value may correspond to fundamentally different confidence levels. So, label-wise uncertainty needs more sufficient model evidence to tell how the model's reasoning process.

### 2.2 DERIVE THE LABEL-WISE TYPE II LIKELIHOOD

In this section, we focus on deriving the Type II likelihood for individual labels. Type II ML Berger & Wolpert (1988) extends Type I by marginalizing over model parameters, effectively integrating out uncertainty in a Bayesian setting. In a fully Bayesian treatment, prior distributions are assigned for the Type I ML given by equation 1 over the hyperparameter $\mu$ and model parameters $\theta$. This makes predictions by marginalizing with respect to the hyperparameter of Type I likelihood and as well as with respect to the parameters $\theta$. In deep neural networks, incorporating hyperpriors for all model parameters Kendall & Gal (2017) remains challenging. Key difficulties include the intractability of directly inferring the posterior distribution of the weights given the data and the high computational cost associated with sampling during inference Amini et al. (2020). Hence in our evidential approximation, we obtain the predictive distribution by introducing hyperpriors over hyperparameters of Type I likelihood. Specifically, due to the limited expressiveness of the Type I likelihood, which may provide insufficient information for the model, we consider defining a prior conjugate Beta distribution on $\mu$. The Beta distribution is expressed as:

$$\mathcal{B}(\mu; a, b) = \frac{\mu^{a-1}(1 - \mu)^{b-1}}{\text{Beta}(a, b)},$$

where $a$ and $b$ are positive hyperparameters. Beta$(a, b)$ is the Beta function Kotz et al. (2019). Then the predictive distribution is obtained by marginalizing over hyperparameters so that:

$$p(t \mid \mathbf{x}, \mathcal{D}) = \int \int p(t \mid \mu)p(\mu \mid a, b)p(a, b \mid \mathbf{x}, \mathcal{D}) \, d\mu d(a, b), \tag{2}$$

where $\mu$, $a$ and $b$ are treated as latent variables and $\mu$ is governed by the $a$ and $b$. $t$ represents the target label of a new given input $\mathbf{x}$. Here, $p(t \mid \mu)$ follows a Bernoulli distribution for a single label, while $p(\mu \mid a, b)$ represents the conjugate Beta prior over $\mu$. The term $p(a, b \mid \mathbf{x}, \mathcal{D})$ denotes the posterior distribution of the hyperparameters $a$ and $b$ given the input $\mathbf{x}$ and the observed data $\mathcal{D}$, encapsulating the model's learned uncertainty about the underlying label probabilities.

**Assumption 1** (Highly Peaked Hyperparameters). *The posterior distribution $p(a, b \mid \mathcal{D})$ is sharply peaked around its mode $(a^*, b^*)$.*

It assumes that the distribution places most of its mass around a specific value. This allows us to approximate the posterior by a delta function centered at $(a^*, b^*)$, i.e., $p(a, b \mid \mathcal{D}) \approx \delta(a - a^*)\,\delta(b - b^*)$ enabling a simpler estimation of the predictive distribution, since exploring the underlying distribution of $a$ and $b$ is much more complex. With Assumption 1, for frequent labels with many matching examples, the (Empirical Bayes) mode of $\mu$ will indeed sharpen, but for rare labels or small datasets, the mode of $\mu$ may sit atop a broad distribution. Now the predictive distribution is derived by integrating over $\mu$, where $a$ and $b$ are set to their sharply peaked values $a^*$ and $b^*$, leading to:

$$p(t \mid \mathbf{x}, \mathcal{D}) \simeq p(t \mid \mathcal{D}, a^*, b^*) = \int p(t \mid \mu)p(\mu \mid \mathcal{D}, a^*, b^*) \, d\mu.$$

To learn the $a^*$ and $b^*$, from Bayes' theorem, the posterior distribution for $a$ and $b$ is expressed as:

$$p(a, b \mid \mathcal{D}) \propto p(\mathcal{D} \mid a, b)p(a, b).$$

**Assumption 2** (Flat Prior $p(a, b)$). *The prior distribution $p(a, b)$ is relatively flat (non-informative).*

The Assumption 2 means it does not favor any particular value of $a$ and $b$ a prior, but a relatively uninformative prior compared with the data, which anchors the Empirical Bayes interpretation. This allows the posterior $p(a, b \mid \mathcal{D})$ to be determined by $p(\mathcal{D} \mid a, b)$, simplifying the estimation of $a^*$ and $b^*$ by maximizing the Type II likelihood $p(\mathcal{D} \mid a, b)$, which is defined as a marginal distribution:

$$p(\mathcal{D} \mid a, b) = \prod_{i=1}^{N} \prod_{k=1}^{K} \int_0^1 p(y_{ik} \mid \mu_{ik})p(\mu_{ik} \mid \mathbf{x}_i, a_{ik}, b_{ik}) \, d\mu_{ik}.$$

The Beta prior distribution, combined with the Bernoulli likelihood, enables the computation of a label-wise marginal likelihood. The Type II likelihood for one single label is expressed as:

$$p(y \mid a, b) = \int_0^1 \mu^y (1 - \mu)^{1-y} \frac{\mu^{a-1}(1 - \mu)^{b-1}}{\text{Beta}(a, b)} \, d\mu.$$

As the inference is shown in Appendix B, the above formula results in:

$$p(y \mid a, b) = \left(\frac{a}{a + b}\right)^y \left(\frac{b}{a + b}\right)^{1-y}. \tag{3}$$

We can also understand that each observed target label $y$ is drawn independently but not identically from the distribution parameterized by $a$ and $b$. These parameters encapsulate our belief about the distribution of labels and are subject to estimation. The graph model in Figure 2 shows the relationship between variables.

## 2.3 MAXIMIZE THE TYPE II LIKELIHOOD BY NEURAL NETWORK

The Type II binary cross-entropy loss is derived from the Type II likelihood in equation 3 by taking a negative logarithm. We approximate second-order hyperparameters $a$ and $b$ by $a(\mathbf{x}, \theta)$ and $b(\mathbf{x}, \theta)$. The loss function is defined as:

$$\mathcal{L}_{TypeII}(f(\theta)) = \log(a(\mathbf{x}, \theta) + b(\mathbf{x}, \theta)) - y\log(a(\mathbf{x}, \theta)) - (1 - y)\log(b(\mathbf{x}, \theta)). \tag{4}$$

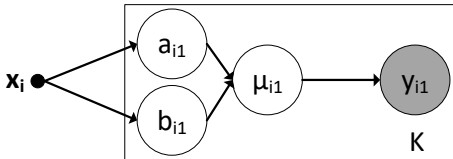

Figure 2: The graph model illustrates building a Beta prior to control the Bernoulli distribution, treating $\mu$ as a latent variable determined by $a$ and $b$, enabling label-wise uncertainty estimation for the number of $K$ labels.

The hyperparameters parameterize the Beta prior $\mathcal{B}(\mu; a, b)$ and are estimated via Empirical Bayes Robbins (1992) by maximizing the marginal likelihood $p(\mathcal{D} \mid a, b)$.[1] The loss function, defined in equation 4, penalizes the model based on its predictive confidence for the label $y$, with $a(\mathbf{x}, \theta)$ and $b(\mathbf{x}, \theta)$ dynamically adjusting to balance the estimated probability and associated uncertainty.

In neural network, we use $\hat{a}_{ik}$ and $\hat{b}_{ik}$ represent the raw logit from the neural network $f(\mathbf{x}_i, \theta)$ of the input $\mathbf{x}_i$ and $k$-th label. We adopt the *softplus* function to ensure positivity and numerical stability during learning. The softplus function $s(x)$ is defined as: $s(x) = \ln(1 + e^x)$, where $s(x) > 0$ for all $x$. Then $a_{ik}$ and $b_{ik}$ can be represented by $s(\hat{a}_{ik})$ and $s(\hat{b}_{ik})$. To generalize this learning process across a dataset with $N$ samples and $K$ classes, we define the empirical risk as Type II binary cross-entropy loss denoted as $\hat{\mathcal{R}}_{\mathcal{D}}^{\text{Type II}}$:

$$\hat{\mathcal{R}}_{\mathcal{D}}^{\text{Type II}}(f(\theta)) = \frac{1}{NK} \sum_{i=1}^{N} \sum_{k=1}^{K} \left[ \log(s(\hat{a}_{ik}) + s(\hat{b}_{ik})) - y_{ik} \log(s(\hat{a}_{ik})) - (1 - y_{ik}) \ln(s(\hat{b}_{ik})) \right].$$

## 2.4 BAYESIAN CLASSIFIER OR GIBBS CLASSIFIER

In Section 2.2 and Section 2.3, we suggested a Bayesian classifier by the Type II likelihood. The loss function in equation 4 is actually a Bayesian classifier, which is given by:

$$\mathcal{L}_{\text{Type II}} = - \log \mathbb{E}_{p(y_{ik}) \sim \text{Beta}(a_{ik}, b_{ik})}[p(y_{ik})],$$

where $p(y_{ik})$ represents the probability of the $k$-th class for the $i$-th sample, and $p(y_{ik}) \sim \text{Beta}(a_{ik}, b_{ik})$ is drawn from a Beta distribution parameterized by $a_{ik}(\mathbf{x}_i, \theta)$ and $b_{ik}(\mathbf{x}_i, \theta)$, which depend on input feature $\mathbf{x}_i$ and the model parameters $\theta$.

In contrast, the Gibbs classifier adopts a different approach. It randomly samples $p(y_{ik})$ from the same Beta distribution, $\text{Beta}(a_{ik}, b_{ik})$, and computes the expected loss by averaging the negative log-probability over these samples. The loss function for the Gibbs classifier, adopted in Zhao et al. (2023), can be defined:

$$\mathcal{L}_{\text{Gibbs}} = \mathbb{E}_{p(y_{ik}) \sim \text{Beta}(a_{ik}, b_{ik})} \left[ - \log p(y_{ik}) \right].$$

Unlike the Bayesian classifier, the Gibbs classifier does not maximize the Type II likelihood but instead relies on Monte Carlo sampling to approximate the expected loss.

To compare the two classifiers, we leverage the convexity of the negative logarithm function. By Jensen's inequality, for a convex function $g$, we have: $g(\mathbb{E}[X]) \leq \mathbb{E}[g(X)]$, which corresponds to $\mathcal{L}_{\text{Type II}} \leq \mathcal{L}_{\text{Gibbs}}$. Equality holds if and only if $g$ is linear over the $p(y_{ik})$ or $p(y_{ik})$ is deterministic (i.e., the Beta distribution is degenerate, with all mass at a single point). Since the negative logarithm is convex, equality requires $p(y_{ik})$ to have zero variance, which occurs only in the degenerate case. Applying to $g(p(y_{ik})) = - \log p(y_{ik})$ and $X = p(y_{ik}) \sim \text{Beta}(a_{ik}, b_{ik})$, we obtain:

$$- \log \mathbb{E}_{p(y_{ik}) \sim \text{Beta}(a_{ik}, b_{ik})}[p(y_{ik})] \leq \mathbb{E}_{p(y_{ik}) \sim \text{Beta}(a_{ik}, b_{ik})} \left[ - \log p(y_{ik}) \right].$$

The inequality implies that, under the same distributional assumption $p(y_{ik}) \sim \text{Beta}(a_{ik}, b_{ik})$, the Bayesian classifier yields a lower or equal risk compared to the Gibbs classifier. Consequently, the Bayesian classifier is more effective as a learning criterion to find the maximum marginal likelihood, as it minimizes a tighter bound on the expected loss. The following proposition formalizes this result.

---

[1]The prior $\mathcal{B}(\mu; a, b)$ is an empirical prior, as its parameters $a$ and $b$ are learned by maximizing the marginal likelihood $p(\mathcal{D} \mid a, b)$.

**Proposition 1** (Bayesian Classifier is More Data-driven). *Let $p(y_{ik}) \sim Beta(a_{ik}, b_{ik})$ represent the class probability for the $i$-th sample and $k$-th class, parameterized by $a_{ik}(\mathbf{x}, \theta)$ and $b_{ik}(\mathbf{x}, \theta)$. Define the Bayesian classifier loss as $\mathcal{L}_{\text{Type II}} = -\log \mathbb{E}_{p(y_{ik}) \sim Beta(a_{ik}, b_{ik})}[p(y_{ik})]$ and the Gibbs classifier loss as $\mathcal{L}_{\text{Gibbs}} = \mathbb{E}_{p(y_{ik}) \sim Beta(a_{ik}, b_{ik})}[-\log p(y_{ik})]$. Then, by Jensen's inequality, $\mathcal{L}_{\text{Type II}} \leq \mathcal{L}_{\text{Gibbs}}$, with equality if and only if $p(y_{ik})$ is deterministic (i.e., the Beta distribution is degenerate). Thus, under the assumption that $p(y_{ik}) \sim Beta(a_{ik}, b_{ik})$, the Bayesian classifier is at least as effective as the Gibbs classifier as a learning criterion to maximum the data likelihood, and more effective when the Beta distribution has non-zero variance.*

Additional experimental results on real-world datasets to validate Proposition 1 between different learning criteria, including the Bayesian classifier (Type II) and the Gibbs classifier, are provided in Appendix C. We also discussed the effect of a non-informative prior weight from Subject Logistic Josang et al. (2018); Jøsang (2016), which is popularly used in EDL Sensoy et al. (2018); Charpentier et al. (2020); Zhao et al. (2023); Bengs et al. (2022). It will make the Beta distribution a Bell shape when we set $a > 1$ and $b > 1$. This can also be found in Appendix C.

## 2.5 Uncertainty Estimation and decomposition

As Proposition 1 demonstrates, the learning criterion of $\mathcal{L}_{\text{Type II}}$ achieves a data likelihood that is greater than or equal to that of the Gibbs classifier. This superior data likelihood indicates that the Type II learning enables the model to develop a more compact connection with the underlying patterns present in the training data. Consequently, the Bayesian classifier's enhanced understanding of training data is crucial for inferring human-interpretable uncertainty measurements. This framework also allows us to model label-wise heteroscedastic uncertainty, contrasting with homoscedastic uncertainty, which assumes a uniform uncertainty level across all data instances.

According to the bias-variance decomposition, we deconstruct the expected variance of $t$ into:

$$E_{\mu \sim (\mu|a,b)} Var_{t \sim p(t|x,\mu,\mathcal{D})}(t) = \underbrace{\frac{ab}{(a+b)(a+b+1)}}_{\textit{Model Variance}} + \underbrace{\frac{ab}{(a+b)^2(a+b+1)}}_{\textit{Model Bias}} + \textit{Noise} \quad (5)$$

where the *Noise* represents the irreducible noise from the training data $D$ and given input $\mathbf{x}$. The *Model Variance* is the expected variance of the predictive distribution on $\mu$. It represents the consistency of the output with respect to the parameters of the current model. The *Model Bias* refers to the discrepancy between the current model and the optimal model given the training data $D$ and input $\mathbf{x}$. The higher the Model Bias, the less knowledge the current model has. All uncertainties are heteroscedastic uncertainties across different labels for each input, derived in Appendix D.1. More introduction about the model bias we defined can be found in Appendix D.3. And according to the Appendix D.2, the total uncertainty of the model can be defined by:

$$\text{Var}_{p(t|a,b)}(t) = \frac{ab}{(a+b)^2} = \textit{Model Variance} + \textit{Model Bias}$$

The total uncertainty of the model primarily depends on the true underlying probability $\mu^* = \frac{a^*(\mathbf{x}, \mathcal{D})}{a^*(\mathbf{x}, \mathcal{D}) + b^*(\mathbf{x}, \mathcal{D})}$, estimated by $\frac{a(\mathbf{x}, \theta, \mathcal{D})}{a(\mathbf{x}, \theta, \mathcal{D}) + b(\mathbf{x}, \theta, \mathcal{D})}$. Under the Robbins-Monro conditions on step sizes, stochastic gradient descent converges to $\mu^*$, which means the total uncertainty of the model $\text{Var}_{p(t|a,b)}(t)$ converges to a certainty level. The proof is deferred to Appendix E.1.

**Proposition 2** (The Total Uncertainty of the Model Will Stabilize with More Data). *As the model learns more underlying patterns of data, the total uncertainty of the model will converge towards a level dictated by the inherent variability of the model parameters.*

In a standard Bayesian framework with a $Beta(a, b)$ prior, observing a new data point $y$ leads to a posterior distribution $p(\mu|y, a, b) \propto p(y|\mu)p(\mu|a, b)/p(y|a, b)$, which results in a Beta posterior with updated parameters: $Beta(a + 1, b)$ if $y = 1$, and $Beta(a, b + 1)$ if $y = 0$. This demonstrates that with each consistent observation, the parameters of the posterior increase, indicating a stronger and more concentrated belief about the underlying parameter $\mu$.

While the neural network $f(\theta)$ does not perform a direct integer update on $a(\mathbf{x}, \theta)$ and $b(\mathbf{x}, \theta)$, the principle of Bayesian updating still guides its learning. When a new data point $(\mathbf{x}, y)$ is observed

consistent with the learned patterns, the neural network will adjust its weights $\theta$ to perform an analogous update to the prior for that input $\mathbf{x}$. This update can be conceptually represented as:

$$p(\mu|y, a(\mathbf{x}, \theta), b(\mathbf{x}, \theta)) \propto p(y|\mu)p(\mu|a(\mathbf{x}, \theta), b(\mathbf{x}, \theta))/p(y|a(\mathbf{x}, \theta), b(\mathbf{x}, \theta))$$

Leading to a posterior that resembles a Beta distribution with updated hyperparameters: Beta($a(\mathbf{x}, \theta) + \delta_a(\mathbf{x}, y), b(\mathbf{x}, \theta)$) if $y$ reinforces the learned pattern, or Beta($a(\mathbf{x}, \theta), b(\mathbf{x}, \theta) + \delta_b(\mathbf{x}, y)$) if $y$ contradicts it (though in the case of consistently patterned data, the reinforcement scenario will be dominant over time). Here, $\delta_a(\mathbf{x}, y) \geq 0$ and $\delta_b(\mathbf{x}, y) \geq 0$ represent the positive adjustments to the respective hyperparameters based on the consistency of the new data $y$ with the learned pattern for input $\mathbf{x}$.

**Assumption 3** (Positive Increasing Hyperparameters $a$ and $b$). *Assume a neural network learns to recognize consistent patterns in a dataset, mapping input features $\mathbf{x}$ to Beta prior hyperparameters $a(\mathbf{x}, \theta)$ and $b(\mathbf{x}, \theta)$ by maximizing the marginal likelihood. As the network observes consistently patterned data points $(\mathbf{x}, y)$, it refines weights $\theta$ to produce positive adjustments $\delta_a(\mathbf{x}, y) \geq 0$ and $\delta_b(\mathbf{x}, y) \geq 0$, leading to increasing values of $a(\mathbf{x}, \theta)$ and $b(\mathbf{x}, \theta)$. This reflects a strengthening belief in the predictive distribution's parameters, analogous to Bayesian posterior updates.*

The neural network's objective is to maximize the marginal likelihood over the training data. For consistently patterned data, the marginal likelihood will be higher for prior hyperparameters $a(\mathbf{x}, \theta)$ and $b(\mathbf{x}, \theta)$ that result in a concentrated prior accurately predicting the outcomes for specific input features. Over increasing amounts of consistent data, the neural network will continuously refine its weights $\theta$ to increase the values of $a(\mathbf{x}, \theta)$ and $b(\mathbf{x}, \theta)$ for those inputs. This increase signifies a growing confidence in the learned prior and a reduction in the model bias for those well-understood input patterns. The proportion of the model bias in the total uncertainty of the model can reflect it:

$$\text{Var}_{p(t|a,b)}(t) = \textit{Model Variance} + \textit{Model Bias} = \textit{Model Variance}(1 + \frac{1}{a+b})$$

where with the increase of the $a + b$, the proportion of the model bias in the total uncertainty is increasing. It shows that the model bias is the dominant term of the total uncertainty of the model when $a + b < 1$, vice versa. Specifically, according to the validation experiments on the real-world dataset in Appendix E.2, the model bias will decrease with more and more training data samples, reflecting reduced model bias in the label uncertainty due to increased confidence in the model's learned prior. Formally, it is specified as follows:

**Proposition 3** (The Model Bias Will Decrease with More Data). *Under the aforementioned Assumption 3, with an increase in consistently patterned data and for the sample size $N$, the model bias $Var_{\mu \sim p(\mu|a(\mathbf{x}, \theta, \mathcal{D}^{(N)}), b(\mathbf{x}, \theta, \mathcal{D}^{(N)}))}[\mu]$ has a higher proportion within the total uncertainty of the model compared to $Var_{\mu \sim p(\mu|a(\mathbf{x}, \theta, \mathcal{D}^{(N+1)}), b(\mathbf{x}, \theta, \mathcal{D}^{(N+1)}))}[\mu]$.*

Interestingly, when we discuss the Beta shape during learning in Section 2.4, we set $a, b > 1$ as a comparison method in Appendix C. It defaults to a relatively smaller proportion of model bias within the total uncertainty of the model, and it does not perform well during training, which implies keep the model thirsty (a high proportion of model bias) could help the learning process.

## 3 EXPERIMENTS

We present a series of experiments to illustrate the effectiveness of our methodology. The results demonstrate that the model bias by uncertainty decomposition offers deeper and more informative insights into the model's reasoning process, enhancing the interpretability. Furthermore, we assess our method using OOD experiments, showcasing its capability to achieve a more reliable and well-calibrated OOD score. All the experiments are conducted on the NVIDIA RTX A4500.

### 3.1 ON THE REAL-WORLD DATASET

To provide an intuitive understanding of label-specific model reasoning on real-world data, we begin by showcasing a detailed result for a representative input. This example elucidates how the model bias offers insights into the potential ambiguities at the individual label level.

This experiment was conducted with a batch size of 16 and used Stochastic Gradient Descent (SGD) as the optimizer with a momentum of 0.9. The initial learning rates are set to $1.5 \times 10^{-4}$ for the

ResNet backbone and $5 \times 10^{-2}$ for the fully connected layer. A Cosine Annealing learning rate scheduler with $T_{\max} = 12$ is applied to adjust the learning rate smoothly over 30 training epochs. The experiment used the Pascal VOC 2012 dataset and was conducted by Resnet-34 He et al. (2016). We compared our method with the energy-based method Wang et al. (2021) and Multi-label evidential learning (MULE) Zhao et al. (2023). The Figure 3, Figure 4 and Figure 5 show three different multi-label uncertainty measurements. Our label-wise model bias provides a human-interpretable reasoning process, revealing how the model weighs visual evidence and highlights ambiguities, thereby enhancing reliability in multi-label classification tasks. More comparative examples are available in Appendix F.1. And we also shows more examples about how the distributional label-wise uncertainty explains the model's thinking in Appendix F.2.

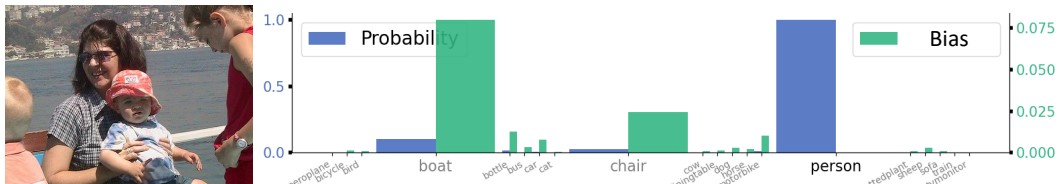

Figure 3: For the ground truth label "person," the model bias is low, indicating high certainty, consistent with the clear depiction of a woman sitting on a chair in an open area. In contrast, higher model bias is observed for the non-ground-truth labels "boat" and "chair," suggesting a lack of knowledge. This aligns with the scene, where multiple boats are visible across the river and a partial chair appears near the woman, potentially confusing the model.

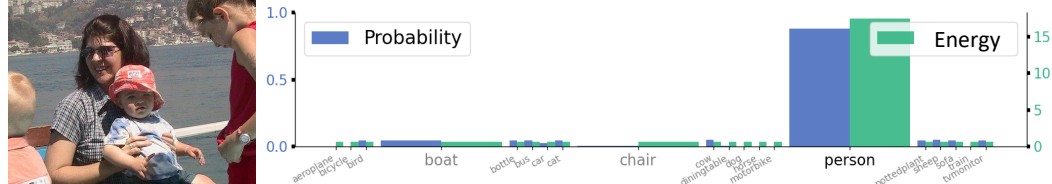

Figure 4: The energy-based method Wang et al. (2021) focuses on the high-probability "person" label, incorrectly indicating uncertainty despite its clear presence in the image.

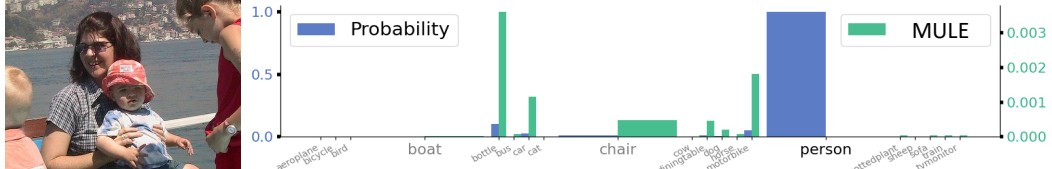

Figure 5: While MULE Zhao et al. (2023) demonstrates certainty for the existence of the ground truth "person" label, the nature of the captured uncertainty for other labels remains less interpretable.

## 3.2 OOD EXPERIMENTS

To further show the statistical effectiveness of our method, we examine OOD experiments for multilabel classification. OOD detection for multi-label classification aims to identify whether an input belongs to the in-distribution $\mathcal{D}_{\text{in}}$ or the out-of-distribution $\mathcal{D}_{\text{out}}$. An input is considered an OOD if it does not contain any label in the in-distribution data. Inputs from $\mathcal{D}_{\text{out}}$ lack labels present in $\mathcal{D}_{\text{in}}$, often representing anomalies encountered during deployment. Consequently, classification models are unable to classify an object as none of the known categories. Instead, they tend to be overconfident in predictions, for example, assigning high probability to previously unseen objects Hendrycks et al. (2019). We can interpret the Beta distribution hyperparameter $a$ as the predicted belief mass. In an out-of-distribution (OOD) scenario, a high value of $a$ may indicate an overconfident prediction, making it a useful signal for OOD detection. A straightforward approach is to compute the sum of the hyperparameter $a$ across all labels as the OOD score, expressed as $\sum_{k=1}^{K} a_{ik}$. The OOD experiments result of this approach is presented as Evidence$_1$ in Table 2.

Table 1: OOD detection performance comparison across different datasets ($\mathcal{D}_{in}$). The results are reported in terms of FPR95 ($\downarrow$), AUROC ($\uparrow$), and AUPR ($\uparrow$). Lower FPR95 and higher AUROC/AUPR indicate better performance. Our proposed methods, **Evidence**$_1$ and **Evidence**$_2$, leverage well-calibrated evidential outputs, achieving superior OOD detection. Notably, incorporating Beta uncertainty in **Evidence**$_2$ significantly reduces FPR95, demonstrating improved reliability.

| $\mathcal{D}_{in}$ | MS-COCO | PASCAL-VOC | NUS-WIDE |
|---|---|---|---|
| **OOD SCORE** | | FPR95$\downarrow$/AUROC$\uparrow$/AUPR$\uparrow$ | |
| MAXLOGIT | 43.53 / 89.11 / 93.74 | 45.06 / 89.22 / 83.14 | 56.46 / 83.58 / 94.32 |
| MSP | 79.90 / 73.70 / 85.37 | 74.05 / 79.32 / 72.54 | 88.50 / 60.81 / 87.00 |
| ODIN | 43.53 / 89.11 / 93.74 | 45.06 / 89.22 / 83.16 | 56.46 / 83.58 / 94.32 |
| MAHALANOBIS | 46.86 / 88.59 / 93.85 | 41.74 / 88.65 / 81.12 | 62.67 / 84.02 / 95.25 |
| LOF | 80.44 / 73.95 / 86.01 | 86.34 / 69.21 / 58.93 | 85.21 / 67.75 / 89.61 |
| ISOLATION FOREST | 94.39 / 49.04 / 66.87 | 93.22 / 50.67 / 35.78 | 95.69 / 53.12 / 83.32 |
| JOINTENERGY | 33.48 / **92.70** / **96.25** | 41.01 / 91.10 / 86.33 | 48.98 / 88.30 / 96.40 |
| **EVIDENCE**$_1$ | 34.14 / 91.08 / 95.14 | 39.28 / 91.67 / 86.38 | 44.82 / **88.57** / **96.53** |
| **EVIDENCE**$_2$ | **29.30** / 92.67 / 96.04 | **37.35** / **92.13** / **88.40** | **42.10** / 86.35 / 95.74 |

Besides, we propose another OOD score that focuses on identifying labels with both a high predicted belief mass and a high model bias. It means a high belief mass, but a lack of knowledge is more like an overconfidence label. To achieve this, we incorporate the confidence associated with each label, represented by the model bias from equation 5. Our objective is to detect labels with an overconfident predicted belief mass while accounting for their uncertainty. A higher score corresponds to a higher probability that the sample is OOD. It can be defined as:

$$\text{Evidence}_2 = \sum_{k=1}^{K} a_{ik} \frac{\text{Var}_{p(\mu|\mathbf{x},\mathcal{D})}(\mu_{ik})}{\sum_{j=1}^{K} \text{Var}_{p(\mu|\mathbf{x},\mathcal{D})}(\mu_{ik})}.$$

Table 2 shows our OOD results by using evidence and the experiment setting follows the Wang et al. (2021). The baselines are Maxlogit Hendrycks et al. (2019), MSP Hendrycks & Gimpel (2016), ODIN Liang et al. (2017), Mahalanobis Lee et al. (2018), LOF Breunig et al. (2000), Isolation Forest Liu et al. (2008) and Joint Energy Wang et al. (2021). The backbone used is DenseNet-121, pretrained on ImageNet-1K Deng et al. (2009). For the in-distribution dataset $\mathcal{D}_{in}$, we use MS-COCO Lin et al. (2014), PASCAL-VOC Everingham et al. (2015), and NUS-WIDE Chua et al. (2009). For the out-of-distribution dataset $\mathcal{D}_{out}$, we select the same set of 20 classes as in Hendrycks et al. (2019) chosen from ImageNet-22K. These classes are specifically selected to avoid overlap with ImageNet-1K, as the multi-label classifiers are pretrained on ImageNet-1K.

From Table 2, we can see that the Evidential training method shows a better OOD performance. When using the OOD score defined as Evidence$_2$, the results demonstrate a significant reduction in the false positive rate at $95\%$ true positive rate (FPR95) compared to others. Our method using Evidence$_2$ achieves a $6.88\%$ lower FPR95 score than the second-best Joint Energy on the NUS-WIDE dataset. This improvement can be attributed to the incorporation of the model bias by the decomposition.

## 4 CONCLUSION AND LIMITATION

Our work introduces a novel perspective for addressing label-wise uncertainty in multi-label classification tasks. In our experiments, we primarily leverage model bias to demonstrate its efficacy; however, we believe its utility extends far beyond this application. Specifically, it holds significant potential to enhance the model's learning process. Furthermore, while model variance and data noise remain underexplored in the literature, we anticipate that they, too, offer substantial opportunities for future research. LLM is used only for polishing specific sentences.

However, in Section 2.5, we give an understandable and heuristic Bayesian update rule explanation in our hierarchical model based on the data consistency, which is conceptual and complex in a deep neural network. More effort on the scalability of data consistency is needed to analyze, for example, how the similarity of data embedding can be considered consistent.

## AUTHOR CONTRIBUTIONS

This work presents a hierarchical Bayesian model for multi-label classification, leveraging a Type II likelihood and Empirical Bayes to achieve robust uncertainty quantification. We theoretically prove the correctness as a data-centric learning criterion and label-wise uncertainty decomposition, demonstrating their alignment with data-driven principles. Amid recent debates on evidential learning, our method outperforms traditional approaches by capturing fine-grained, human-interpretable model bias. The intuitive experiments show that deep neural networks can have human-like thinking of uncertainty when facing complex objects, and this can be explained by label-wise uncertainty deconposition defined in equation 5. In out-of-distribution (OOD) detection, our method achieves up to a $6.88\%$ lower FPR95 score compared to the second-best baseline on a benchmark dataset, showcasing its practical efficacy.

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

## A    TYPE I & II MAXIMUM LIKELIHOOD

Type I ML directly optimizes the likelihood of the observed data with respect to the model parameters. The parameter values are determined directly by maximizing the likelihood $P(\mathcal{D}|\theta)$, where $\mathcal{D}$ is the observed data with ground truth and $\theta$ represents the parameters of model $f$.

Type II ML is an approximation framework, which is also called Evidence Approximation MacKay (1992), setting the hyperparameters $m$ to specific values determined by maximizing the marginal likelihood function obtained by first integrating over the parameters $\theta$. This gives rise to the marginal likelihood as shown:

$$p(\mathcal{D}|m) = \int p(\mathcal{D}|\theta)p(\theta|m)\, d\theta.$$

The marginal likelihood $p(\mathcal{D}|m)$ is a function that describes how likely the observed data $\mathcal{D}$ is, given the parameters $m$. It essentially answers the question: " How well do the parameters $m$ explain the observed data $\mathcal{D}$? " It serves as the foundation for optimization, where the model adjusts $m$ to maximize the agreement between predictions and observed data.

Once the marginal likelihood is derived, the next step involves estimating the hyperparameters $m$ by maximizing the marginal likelihood expression with respect to $m$. This is often referred to as Type II Maximum Likelihood estimation, which seeks to identify the set of hyperparameters $m$ that best explain the observed data under the given model. By maximizing this marginal likelihood, we effectively integrate out latent variables or nuisance parameters, allowing the model to focus solely on optimizing the hyperparameters to enhance its overall fit. The estimation will be:

$$\hat{m} = \arg \max_m p(\mathcal{D}|m).$$

## B    INFERENCE OF TYPE II LIKELIHOOD

The Beta prior distribution, combined with the Bernoulli likelihood, enables the computation of a label-wise marginal likelihood. The marginal likelihood for one single label is expressed as:

$$p(y\,|a,b) = \int_0^1 \mu^y (1-\mu)^{1-y} \frac{\mu^{a-1}(1-\mu)^{b-1}}{\mathrm{Beta}(a,b)}\, d\mu$$

$$= \frac{1}{\mathrm{Beta}(a,b)} \int_0^1 \mu^{y+a-1}(1-\mu)^{1-y+b-1}\, d\mu$$

$$= \frac{\mathrm{Beta}(y+a, 1-y+b)}{\mathrm{Beta}(a,b)}.$$

Here, the Beta function $\mathrm{Beta}(a,b)$ serves as a normalization constant. Using the relationship between the Beta and Gamma functions, where $\mathrm{Beta}(a,b) = \frac{\Gamma(a)\Gamma(b)}{\Gamma(a+b)}$, we rewrite the marginal likelihood as:

$$p(y|a,b) = \frac{\Gamma(y+a)\Gamma(1-y+b)}{\Gamma(1+a+b)} \cdot \frac{\Gamma(a+b)}{\Gamma(a)\Gamma(b)}.$$

To further illustrate this, consider the two possible cases for the binary ground truth $y$:

$$p(y|a,b) = \begin{cases} \frac{\Gamma(a)\Gamma(1+b)}{\Gamma(1+a+b)} \cdot \frac{\Gamma(a+b)}{\Gamma(a)\Gamma(b)}, & \text{if } y = 0, \\ \frac{\Gamma(1+a)\Gamma(b)}{\Gamma(1+a+b)} \cdot \frac{\Gamma(a+b)}{\Gamma(a)\Gamma(b)}, & \text{if } y = 1. \end{cases}$$

Finally, leveraging the property of the Gamma function, $\Gamma(x+1) = x\Gamma(x)$, we simplify the above expression into the Type II Maximum Likelihood (Type II ML) form. This results in the evidential likelihood:

$$p(y|a,b) = \left(\frac{a}{a+b}\right)^y \left(\frac{b}{a+b}\right)^{1-y}.$$

Table 2: Average Precision scores for different loss functions on ResNet architectures.

| LOSS FUNCTION | RESNET-34 | RESNET-50 |
|---|---|---|
| $\mathcal{L}_{\text{BCE}}$ | 0.922 | 0.938 |
| $\mathcal{L}_{\text{BCE}}(\text{SOFTMAX})$ | 0.918 | 0.927 |
| $\mathcal{L}_{\text{TYPE II}}(a, b > 1)$ | 0.929 | 0.940 |
| $\mathcal{L}_{\text{GIBBS}}$ | 0.729 | 0.688 |
| $\mathcal{L}_{\text{TYPE II}}$ | **0.947** | **0.959** |

## C  COMPARISON ON REAL-WORLD DATASET

We compare the average precision score on the Pascal VOC 2012 real-world dataset for the Bayesian classifier $\mathcal{L}_{\text{Type II}}$, the Gibbs classifier $\mathcal{L}_{\text{Gibbs}}$, and other training criteria. The other criteria include binary cross entropy loss $\mathcal{L}_{\text{BCE}}$, softmax binary cross entropy loss $\mathcal{L}_{\text{BCE}}(\text{softmax})$, and $\mathcal{L}_{\text{Type II}}(a, b \geq 1)$. The latter represents the setting where hyperparameters $a$ and $b$ are constrained to be greater than or equal to 1, a common configuration in evidential deep learning. As shown in table 2, the Type II learning criterion achieves superior performance on the dataset.

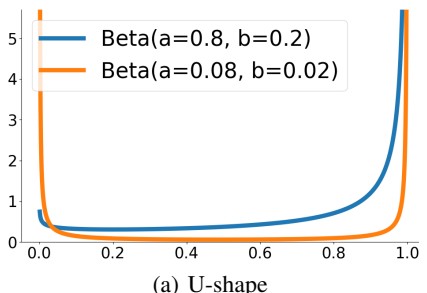

(a) U-shape

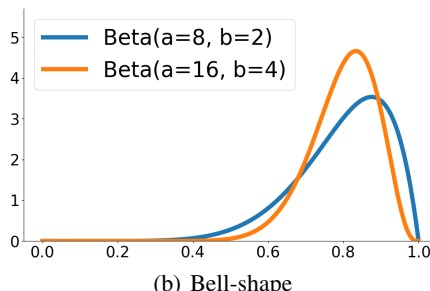

(b) Bell-shape

Figure 6: Given the predicted probability is 80%, different shapes of Beta distribution.

As we can see from Figure 6, when both the hyperparameters $a$ and $b$ of a Beta distribution are less than 1, the distribution exhibits a U-shaped, meaning most of the probability density is concentrated near 0 and 1. Beta distribution in U-shape reflects that the true probability $p$ is likely to be close to one of the extremes.

For $a, b > 1$, the Beta distribution is unimodal (bell shape), with the mode positioned between 0 and 1. The constraint $a, b > 1$ is commonly used for numerical stability and smoother predictions Sensoy et al. (2018); Bao et al. (2021), as it incorporates a non-informative prior greater than 1. However, relaxing this constraint of non-informative prior allows the distribution to take U-shaped or L-shaped forms, reflecting a strong preference for probabilities near 0 and 1. This increased flexibility provides more expressive modeling for Type II ML estimation. The result in table 2 also shows that in the context of binary classification, without the restriction of $a, b > 1$, the Type II maximum likelihood actually can flexibly to adjust the hyperparameters to achieve the maximum and have better performance. Binary classification typically involves clear separations between two classes, making such extreme probabilities more plausible. The U-shaped and L-shaped Beta distributions align with the nature of the task and the available data. This belief suits scenarios where predictions are expected to lean strongly toward one of the two classes.

## D  TWO UNCERTAINTY METHODS

### D.1  UNCERTAINTY DECOMPOSITION

In conventional classification tasks, labels are typically treated as deterministic, being either true or false, which allows uncertainty to be quantified using probability estimates, while the value of

probability often lacks the richness needed to capture the full complexity of uncertainty. However, real-world data is often subject to various sources of noise, such as measurement errors, data scarcity, and model limitations, leading to non-deterministic behavior and incomplete knowledge. In our work, we decompose the uncertainty into:

$$Var_{t \sim p(t|x,\mu,\mathcal{D})}(t) = E_t[(p(t=1|x,\mu,\mathcal{D}) - t)^2]$$

$$= E_t[(p(t=1|x,\mu,\mathcal{D}) - E_t[p(t=1|x,\mathcal{D})])^2] + E_t[(E_t[p(t=1|x,\mathcal{D})] - t)^2]$$

$$E_{\mu \sim (\mu|a,b)} Var_{t \sim p(t|x,\mu,\mathcal{D})}(t) = E_t[(E_t[p(t=1|x,\mathcal{D})] - t)^2] +$$

$$+ E_{\mu \sim (\mu|a,b)} \{ [p(t=1|x,\mu,\mathcal{D}) - E_{\mu \sim (\mu|a,b)}[p(t=1|x,\mu,\mathcal{D})]]^2$$

$$+ [E_{\mu \sim (\mu|a,b)}[p(t=1|x,\mu,\mathcal{D})] - E_t[p(t=1|x,\mu,\mathcal{D})]]^2 \}$$

$$= \underbrace{E_t[(E_t[p(t=1|x,\mathcal{D})] - t)^2]}_{Noise} +$$

$$\underbrace{E_{\mu \sim (\mu|a,b)} \left\{ E_t \left[ p(t=1|x,\mu,\mathcal{D}) - E_{\mu \sim (\mu|a,b)}[p(t=1|x,\mu,\mathcal{D})] \right]^2 \right\}}_{Model\ Variance} +$$

$$\underbrace{E_{\mu \sim (\mu|a,b)} \left\{ \left[ E_{\mu \sim (\mu|a,b)}[p(t=1|x,\mu,\mathcal{D})] - E_t[p(t=1|x,\mathcal{D})] \right]^2 \right\}}_{Model\ Bias}$$

$$= Noise + E_{\mu \sim p(\mu|a,b)} [Var_t[p(t|x,\mu,\mathcal{D})]] + Var_{\mu \sim p(\mu|a,b)} [E_t [p(t|x,\mu,\mathcal{D})]]$$

$$= Noise + E_{\mu \sim p(\mu|a,b)} [\mu(1-\mu)] + Var_{\mu \sim p(\mu|a,b)} [\mu]$$

$$= Noise + \underbrace{\frac{ab}{(a+b)(a+b+1)}}_{Model\ Variance} + \underbrace{\frac{ab}{(a+b)^2(a+b+1)}}_{Model\ Bias}$$

where the $E_{\mu \sim p(\mu|a,b)} [\mu(1-\mu)]$ can be derived by:

$$E_{\mu \sim p(\mu|a,b)} [\mu(1-\mu)] = E_{\mu \sim p(\mu|a,b)}[\mu] - E_{\mu \sim p(\mu|a,b)}[\mu^2]$$

$$= E_{\mu \sim p(\mu|a,b)}[\mu] - [Var_{\mu \sim p(\mu|a,b)}(\mu) + (E_{\mu \sim p(\mu|a,b)}[\mu])^2]$$

$$= \frac{a}{a+b} - \frac{ab}{(a+b)^2(a+b+1)} - (\frac{a}{a+b})^2$$

$$= \frac{ab}{(a+b)(a+b+1)}$$

### D.2 THE TOTAL MODEL VARIANCE

As we mentioned before, for notational simplicity, we denote these data-driven hyperparameters $a(\mathbf{x}, \theta)$ and $b(\mathbf{x}, \theta)$ as $a$ and $b$. The predictive distribution is approximated by:

$$p(t \mid \mathbf{x}, \mathcal{D}) \approx p(t \mid a, b) = \int_0^1 p(t \mid \mu) \text{Beta}(\mu \mid a, b) \, d\mu.$$

where the approximation of the predictive distribution does not consider the noise of $\mathbf{x}$ and $\mathcal{D}$, since hyperparameters $a$ and $b$ are approximated by $a(\mathbf{x}, \theta)$ and $b(\mathbf{x}, \theta)$, so the total variance of the real predictive distribution is equal to the sum of the total variance of the current model and the noise of $\mathbf{x}$ and $\mathcal{D}$. The total variance of the real predictive distribution can be represented by:

$$\text{Var}_{p(t|\mathbf{x},\mathcal{D})}(t) = E_{p(t|\mathbf{x},\mathcal{D})}[t^2] - \left( E_{p(t|\mathbf{x},\mathcal{D})}[t] \right)^2.$$

Since $t^2 = t$, $E_{p(t|\mathbf{x},\mathcal{D})}[t^2] = E_{p(t|\mathbf{x},\mathcal{D})}[t]$. The mean is:

$$E_{p(t|\mathbf{x},\mathcal{D})}[t] = p(t=1 \mid \mathbf{x}, \mathcal{D}) \approx \int_0^1 \mu \text{Beta}(\mu \mid a, b) \, d\mu = \frac{a}{a+b}.$$

Thus, the total model variance conditioned on the $\mathbf{x}$ and $\mathcal{D}$ can be approximated by:

$$\text{Var}_{p(t|\mathbf{x},\mathcal{D})}(t) \approx \text{Var}_{p(t|a,b)}(t) = \frac{a}{a+b} - \left( \frac{a}{a+b} \right)^2 = \frac{ab}{(a+b)^2} = model\ variance + model\ bias$$

where the approximated model variance is equal to the sum of the model variance and the model bias.

### D.3 The Model Bias

To visualize the behavior of the predictive distribution across different combinations of the Beta distribution hyperparameters $a$ and $b$, we present heatmaps in Figure 7, which illustrate the mean and variance of the predictive distribution $p(\mu \mid \mathbf{x}, \mathcal{D}) \approx \text{Beta}(\mu \mid a, b)$. The left heatmap depicts the mean $\mathbb{E}_{p(\mu \mid \mathbf{x}, \mathcal{D})}[\mu] \approx \frac{a}{a+b}$, representing the predicted probability, while the right heatmap shows the model bias $\text{Var}_{p(\mu \mid \mathbf{x}, \mathcal{D})}(\mu) \approx \frac{ab}{(a+b)^2(a+b+1)}$, a measure of model for the lack of knowledge. These visualizations reveal how the interplay of $a$ and $b$ generates a wide range of probability estimates and uncertainty levels, with variance peaking when $a$ and $b$ are small and balanced (indicating maximum uncertainty) and decreasing as either parameter dominates, reflecting a more confident prediction.

To further explore this distributional uncertainty of model bias, Figure 8 contrasts it with traditional entropy-based uncertainty in a 3D representation. The left subfigure (Figure 8(a)) plots entropy for a binary classification case, where uncertainty remains constant for a fixed probability $p = 0.5$ (maximum entropy), offering a static measure of ignorance. In contrast, the right subfigure (Figure 8(b)) depicts a 3D surface of model bias as a function of $a$ and $b$, highlighting its dynamic nature. Unlike entropy, which provides a uniform uncertainty measure for a given $p$, the Beta distribution's variance varies with $a$ and $b$, enabling a richer representation of uncertainty that captures the model's confidence in the predictive distribution. This variability is particularly valuable in multi-label classification, where label-wise uncertainty (e.g., for a stable person vs. a moving car) can differ significantly due to distinct $a(\mathbf{x}, \theta)$ and $b(\mathbf{x}, \theta)$ inferred by the neural network.

The dependency of variance on $a$ and $b$ aligns with our Empirical Bayes framework, where these hyperparameters are learned by maximizing the Type II likelihood. As consistent data patterns reinforce the model (per Assumption 3), $a(\mathbf{x}, \theta)$ and $b(\mathbf{x}, \theta)$ increase, reducing variance and concentrating the posterior, as observed in the heatmaps. This adaptability offers deeper insights into model confidence, facilitating applications such as out-of-distribution (OOD) detection, where precise uncertainty quantification enhances reliability. Moreover, the 3D visualization underscores the potential for leveraging model bias in safety-critical domains, where understanding label-specific confidence is crucial for interpreting model behavior and ensuring robust predictions.

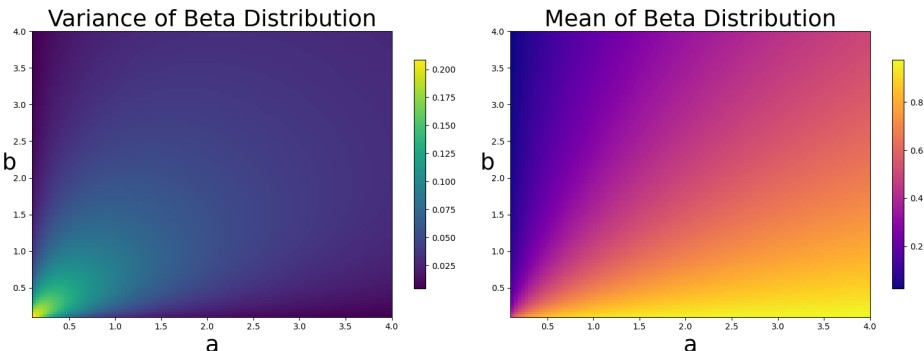

Figure 7: The combination of the parameters $a$ and $b$ in the Beta distribution results in a variety of probabilities and variances.

## E  Behavior of Aleatoric and Epistemic Uncertainty

### E.1  Convergence of Aleatoric Uncertainty

*Proof.* The estimated total uncertainty of the model is given by:

$$\text{Var}_{p(t \mid a(\mathbf{x}, \theta, \mathcal{D}), b(\mathbf{x}, \theta, \mathcal{D}))}(t) = \frac{a(\mathbf{x}, \theta, \mathcal{D}) b(\mathbf{x}, \theta, \mathcal{D})}{(a(\mathbf{x}, \theta, \mathcal{D}) + b(\mathbf{x}, \theta, \mathcal{D}))^2} = \hat{\mu}(1 - \hat{\mu}),$$

where $\hat{\mu}$ is the estimated mean of the predictive distribution of the label.

Maximizing the Type II likelihood $\mathbb{E}_{p(y_{ik}) \sim \text{Beta}(a_{ik}, b_{ik})}[p(y_{ik})]$ does not directly maximize the Type I likelihood $p(y_{ik} \mid \mu)$, because Type II integrates over all possible values of $\mu$ according to the prior.

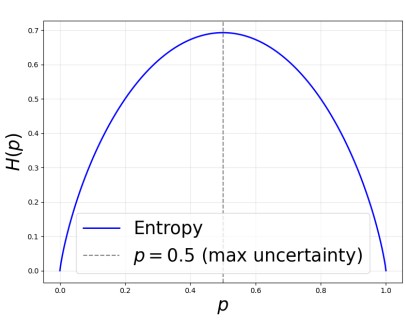 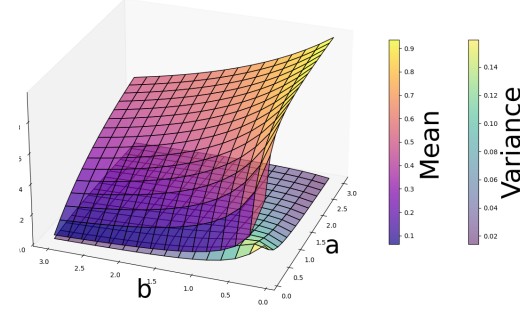

(a) Entropy uncertainty (binary case)    (b) Uncertainty using the variance of Beta distribution

Figure 8: For a given fixed probability, entropy provides a constant measure of uncertainty, whereas the Beta distribution allows for variable uncertainty depending on its hyperparameters, offering deeper insights into model confidence.

However, the outcome of maximizing Type II (finding optimal $a$ and $b$) is a Beta distribution whose mean $\frac{a}{a+b}$ serves as a good point estimate for $\mu$, and this point estimate is the value that would maximize the Type I likelihood for the observed data. The per-sample gradient of the Type-II loss is an unbiased estimate of the gradient of the binary cross-entropy risk. Under the Robbins–Monro conditions on step sizes, SGD then converges to the true parameter. So $\hat{\mu}$ will converge towards the true underlying probability $\mu^*$ of the label associated with the input $\mathbf{x}$. Therefore, the total uncertainty of the model will converge to: $\mu^*(1-\mu^*)$, which represents the inherent Bernoulli variance associated with the consistent underlying pattern. It signifies that even with perfect learning of the pattern, there will still be irreducible randomness in the label according to the underlying likelihood. The total uncertainty of the model will not necessarily converge to zero unless the underlying pattern is deterministic ($\mu^* = 0$ or $\mu^* = 1$). Instead, it converges to a level dictated by the inherent stochasticity of the consistently patterned data. $\qquad\square$

## E.2 DECREASE OF EPISTEMIC UNCERTAINTY

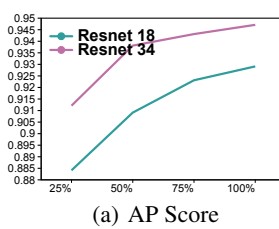 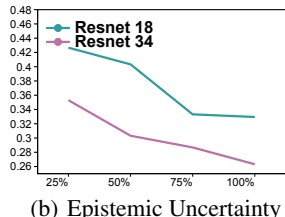 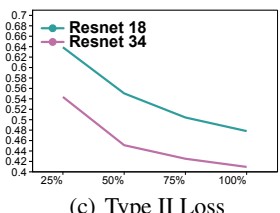

(a) AP Score    (b) Epistemic Uncertainty    (c) Type II Loss

Figure 9: Epistemic Uncertainty Reduction with Increasing Data on Pascal VOC 2012

Figure 9 provides experimental validation for Proposition 3. As the training data incrementally increased from $25\%$ to $100\%$, the average precision (shown in Figure 9(a)) progressively improved while the associated risk (depicted in Figure 9(c)) decreased. Concurrently, the epistemic uncertainty, as quantified in equation 5, exhibited a corresponding reduction. The experimental setting is the same as in Section 3.

# F    INTUITIVE EXPERIMENTS

## F.1    ADDITIONAL COMPARATIVE EXAMPLES

We present further comparative experiments, where blue bars represent the model's predictive probability for each label, and green bars illustrate the corresponding uncertainty as quantified by different methods across figures. The black labels are the ground truth.

The first comparative example, shown in Figure 10, examines a scene containing a dog and chairs, with a very likely dining table. Our method's distributional uncertainty indicates high certainty regarding the obvious presence of the dog. And it reflects the model's bias for the low predictive probability of the dining table. The method also captures a moderate uncertainty for the chairs, which appears reasonable given the potentially ambiguous viewing angle of these objects. In contrast, both the Energy-based method Wang et al. (2021) and Multi-label Evidential Learning (MULE) Zhao et al. (2023) suggest uncertainty for the dog label, a result that lacks clear interpretability based on the visual evidence. In another instance shown in Figure 11, where "cat" and "tvmonitor" are the ground

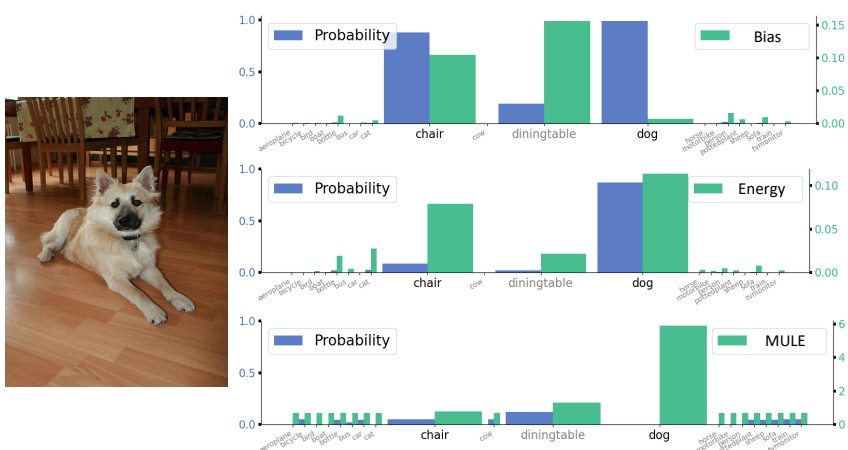

Figure 10: High model bias for the low predictive probability of the dining table.

truth labels, the predictive probabilities of all three models exhibit similar trends: low for "tvmonitor" and high for "cat." However, our method uniquely provides an interpretable uncertainty assessment, indicating certainty for the high probability assigned to "cat" and uncertainty for the low probability associated with "tvmonitor." This aligns with a human understanding of the scene.

## F.2    MORE ANALYSIS ON REAL WORLD DATA

We can examine more results in Figure 12 from the Pascal VOC 2012 dataset conducted by Resnet-34 He et al. (2016). The blue bar represents predictive probability. Meanwhile, the green bar represents the model distributional uncertainty. The black label indicates the ground truth, while the gray labels represent non-ground truth labels. Starting with data ambiguity, several examples demonstrate the challenges within the dataset. In the first image on the left, a potted plant is visible but excluded from the ground truth, resulting in high variance in the model's predictive probability. Similarly, the second image contains a figure resembling the KFC logo or the people on the ground, yet lacks a "person" ground truth label. The third image depicts a piece of furniture that could be classified as either a chair or a sofa, adding inherent ambiguity. The fifth image on the left shows a blurry car in the background, further confusing the model. On the right side, the first image highlights a person behind a cute dog, yet the label "person" is missing from the ground truth, contributing to uncertainty. Turning to the model's limited knowledge, several cases reveal perplexity. For instance, the fourth image on the left features a crashed motorcycle with high label nervousness due to limited prior knowledge. Similarly, the fifth image on the right depicts a sheep being sheared. It is interesting to

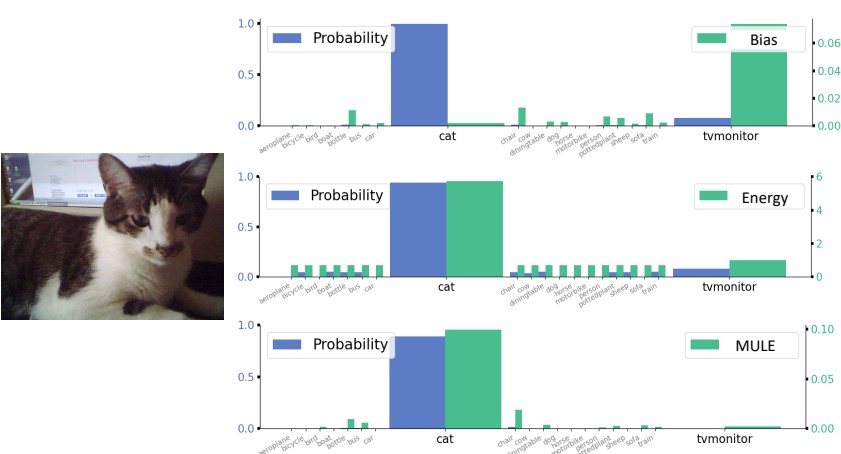

Figure 11: High model bias for the low predictive probability of the TV monitor.

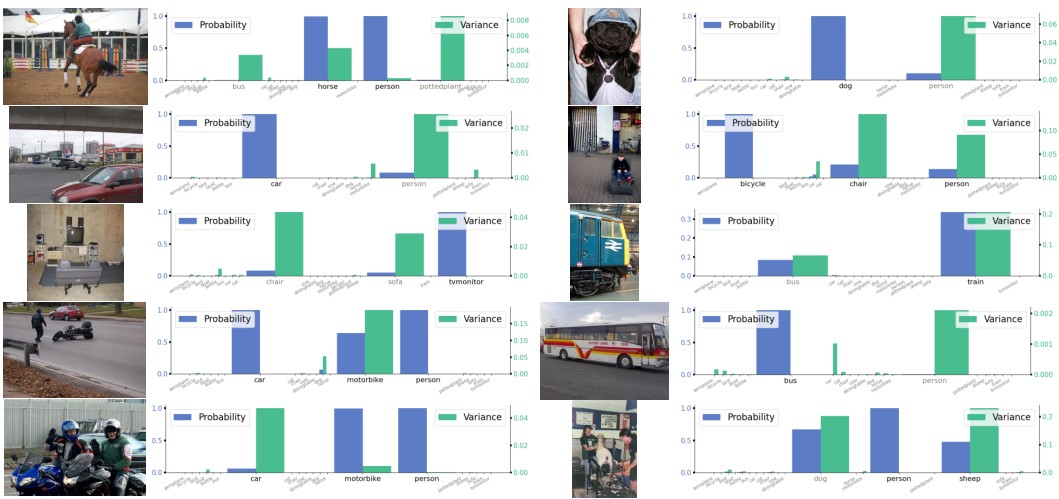

Figure 12: This figure presents real-world data experiment results, showcasing the model's fine-grained label-wise epistemic uncertainty.

discuss the explainable variance levels illustrated in Figure 12, which reflect human-interpretable label-wise epistemic uncertainties. The model's "feeling" can range from confident to confused. These results provide insights into the model's internal reasoning and offer an intuitive answer to the question: "Does the model exhibit 'nervousness' in each label-wise prediction?"

The result shows that our approach enables a deeper and label-wise understanding of the model's predictive probability. Whether grappling with ambiguous ground truths, such as blurry objects or misclassified categories, or facing scenarios outside its learned knowledge, like unusual contexts or rare events, the method highlights areas of uncertainty that impact decision-making. This ability to expose and explain uncertainty provides valuable insights for improving model robustness and interpretability, helping to align predictions with reasoning in real-world applications.

