# OpenReview forum: "Label-Wise uncertainty decomposition for Multi-label Classification by Maximizing Type II Likelihood"
_ICLR.cc/2026/Conference — ICLR 2026 Conference Withdrawn Submission_

### Official Review · Reviewer_uayy · 2025-10-24

**Soundness:** 2
**Presentation:** 2
**Contribution:** 3
**Rating:** 4
**Confidence:** 2

**Summary:**

The paper introduces a Bayesian framework for multi-label classification, leveraging Type II likelihood to estimate label-wise uncertainty. The key idea is to predict Beta-distributed hyperparameters per label, enabling uncertainty decomposition into model variance, bias, and noise via bias–variance analysis. The authors claim interpretability and improved out-of-distribution (OOD) detection using Pascal VOC, COCO, and NUS-WIDE.

**Strengths:**

1. The use of Type II likelihood for multi-label uncertainty quantification is an interesting theoretical contribution that bridges Bayesian evidence approximation and evidential deep learning. The paper provides rigorous derivations linking the marginal likelihood, bias-variance decomposition, and learning criterion.

2. The experimental improvements are empirically meaningful, and the proposed “Evidence2” score is intuitively motivated. Moreover, the qualitative examples are helpful in showing how label-wise uncertainty behaves and supports explanation.

**Weaknesses:**

1. The experiments are relatively narrow in scope — mainly on Pascal VOC and NUS-WIDE — without ablation on different architectures, datasets, or scalability. The performance gains, while interesting, are modest and may not justify the added complexity.

2. Key results depend on strong assumptions (e.g., sharply peaked posterior in Assumption 1, flat prior in Assumption 2). The practical effect of these approximations is not tested or discussed.

**Questions:**

1. How sensitive are results to the Beta hyperparameter initialization or to the network’s scale?

2. How does this method scale to larger label sets (e.g., 100+ labels or open-vocabulary tasks)?

---

### Official Review · Reviewer_6MTF · 2025-10-31

**Soundness:** 2
**Presentation:** 2
**Contribution:** 2
**Rating:** 2
**Confidence:** 4

**Summary:**

This paper introduces a method to model label-wise uncertainty in a multi-label classification setting. This is done by having a neural network learn the parameters of a Beta distribution for each class through optimization of a Type II Likelihood-based cross-entropy loss function. This approach is discussed in terms of propositions that describe the ‘’behaviour’’ of the method during learning, and a bias-variance decomposition. The bias-variance decomposition also gives rise to an uncertainty measure that is used in experiments. Empirically, the proposed method is evaluated by considering the predicted uncertainty for selected examples and by performing out-of-distribution (OOD) detection. In the latter experiment, the models achieves superior performance on two out of the three evaluated datasets.

**Strengths:**

- The paper tackles the important problem of multi-label classification.
- The proposed approach achieves good results on selected examples and in the OOD experiments.

**Weaknesses:**

- There is no mention / results of how the method actually performs in the multi-label classification task. Relevant metrics should be reported.
- The method is only properly evaluated on OOD detection. It would benefit of a more extensive evaluation on different tasks such as, for example, selective prediction.
- The paper does not adequately address how the drawbacks of evidential learning are addressed in the current work.
- While the paper mentions multiple times that the label-wise uncertainty provides valuable insights, it is unclear how these insights can actually improve decision-making. As illustrated in Figure 1, the approach seems to produce meaningful label-wise uncertainty, but it is not discussed how this information can actually be used in a beneficial way.
- No code is provided.
- The figures are not good. For example, Figure 1 is very messy and hard to read. For Figures 3, 4, and 5 it is unclear what the maximum and minimum of the metric are; is e.g., a bias of 0.075 high or low?

**Questions:**

- Could you clarify how your paper addresses the problems with evidential learning, as mentioned in the introduction.
- Why do the OOD measures proposed in Section 3.2 depend only on $a$ and not $b$? If, for example, both $a$ and $b$ are high, the model is actually not confident. Why is this a meaningful measure of uncertainty?
- What concrete benefits does the label-wise uncertainty have in terms of tasks that use uncertainty?

---

### Official Review · Reviewer_QoP1 · 2025-11-01

**Soundness:** 3
**Presentation:** 3
**Contribution:** 2
**Rating:** 4
**Confidence:** 5

**Summary:**

In this paper, the authors propose a hierarchical Bayesian framework for multi-label classification to address the limitations of conventional deep learning models in label-wise uncertainty quantification. Specifically, the authors propose to maximize Type II likelihood with Empirical Bayes and predicts hyperparameters of a Beta prior over Bernoulli label probabilities with a neural network.

The experiments are conducted on several benchmark datasets when the expeirmental results demonstrate improvments based on ResNet backbones. Some visual examples also illustrate how the decomposed uncertainty aligns with human reasoning.

**Strengths:**

* The method is rigorously grounded in Bayesian inference, with proofs showing its superiority as a data-centric learning criterion and convergence properties for uncertainty components.
* The decomposition, especially model bias, provides human-aligned insights into label-specific ambiguities, as evidenced by intuitive visualizations that reveal model "nervousness" in real-world scenarios.
* The method extends evidential deep learning to label-wise heteroscedastic uncertainty without assuming label independence explicitly, outperforming baselines in accuracy and calibration on benchmark datasets.

**Weaknesses:**

* The label dependencies are ignored when the binary relevance approach treats labels independently.
* The method depends on approximations like highly peaked posteriors and flat priors, which may not hold for small or noisy datasets, risking inaccurate uncertainty estimates.
* Maximizing Type II likelihood and decomposing uncertainty may increase training/inference costs compared to standard BCE, though not quantified, limiting scalability for large-scale applications.

**Questions:**

* How does the proposed method perform when applied to datasets with extremely imbalanced label distributions?
* What are the specific computational costs (e.g., training time, memory usage) associated with maximizing Type II likelihood compared to Type I likelihood?
* How does the method handle missing or incomplete label information in the training data?
* How does the model scale with an increasing number of labels or even with the extreme multi-label classification (XMC) tasks?

---

### Official Review · Reviewer_JgQc · 2025-11-02

**Soundness:** 1
**Presentation:** 2
**Contribution:** 2
**Rating:** 2
**Confidence:** 3

**Summary:**

The paper proposes a loss similar to the Dirichlet-based loss commonly used in EDL (evidence-based deep learning).
The proposed method is predicts the parameters of a Beta distribution per-label in multi-label classification setting, which are denoted by 'a', and 'b'. The variance of the Beta distribution is used for uncertainty quantification. Moreover, 'a' value is used for OOD detection.
The final objective for training becomes simply the cross entropy loss on the expected output of the predicted distribution $Beta(a,b)$, i.e. $\frac{a}{a+b}$.

In summary, I'm rating this paper as  '2: reject, not good enough', but I'm open to change my score if the authors properly address my comments and concerns.

**Strengths:**

- The introduction and the review of related work is insightful.
- The quantitative and qualitative evaluations are to some degree convincing.

**Weaknesses:**

- In Sec. 2.4. two losses are compared. First of all, the term 'Gibss classifier' refers to sampling one of predictors and reporting its predictions, instead of marginalising out the belief over predictors. So Gibbs classifier doesn't have any specific training objective, and is used to broadly refer to the afformentioned prediction strategy.
- Proposition 1 is simply the Jenson inequality. Putting that aside, I don't see how general claims like 'develop a more compact connection with the underlying patterns present in the trainning set' can be made from $\mathcal{L}_{Type II} < \mathcal{L})_{Gibbs}$.
- Assumptions 1-3 are somewhat reasonable, but the connection of each assumption to the derivations should be explicitly  mentioned. In other words, I don't see how they get connected to each proposition and derivation in the paper.
- In Sec. 3.2. OOD samples are defined as images to which none of training set labels are assignable. But even in this case, since the setting is multi-label, the model can simply predict all zeros (i.e. no label assigned) with a high confidence if the input image is in-distribution (i.e. in-distribution in its commonly-used meaning) . Indeed, I don't see why the multi-label model has to express high uncertainty (or high 'a' value) for these instance.
- Not included in score:
  - Gramatical issue in line 49: 'While ... . Their method primarily'. The two sentences should be connected.
  - Line 142, vague phrase "deeper levels of uncertainty"
  - Line 189 "The Assmuption 2" ===> "The" is not needed.
  - Line 335, vague phrase "consistently patterned data points"
  - Line 382, "The Figure 2" ===> "The" is not needed

**Questions:**

Please refer to the comments in the 'Weaknesses' section.

---

### Note · Authors · 2025-11-12

I have read and agree with the venue's withdrawal policy on behalf of myself and my co-authors.